# Research on Sensitivity Improvement Methods for RTD Fluxgates Based on Feedback-Driven Stochastic Resonance with PSO

**DOI:** 10.3390/s25020520

**Published:** 2025-01-17

**Authors:** Rui Wang, Na Pang, Haibo Guo, Xu Hu, Guo Li, Fei Li

**Affiliations:** College of Computer Science and Technology, Beihua University, No. 3999 East Binjiang Road, Jilin 132013, China; wangrui@beihua.edu.cn (R.W.); ghb@beihua.edu.cn (H.G.); hx@beihua.edu.cn (X.H.); lg@beihua.edu.cn (G.L.); lifei@beihua.edu.cn (F.L.)

**Keywords:** stochastic resonance, RTD fluxgate, sensitivity, feedback, PSO

## Abstract

With the wide application of Residence Time Difference (RTD) fluxgate sensors in Unmanned Aerial Vehicle (UAV) aeromagnetic measurements, the requirements for their measurement accuracy are increasing. The core characteristics of the RTD fluxgate sensor limit its sensitivity; the high-permeability soft magnetic core is especially easily interfered with by the input noise. In this paper, based on the study of the excitation signal and input noise characteristics, the stochastic resonance is proposed to be realized by adding feedback by taking advantage of the high hysteresis loop rectangular ratio, low coercivity and bistability characteristics of the soft magnetic material core. Simulink is used to construct the sensor model of odd polynomial feedback control, and the Particle Swarm Optimization (PSO) algorithm is used to optimize the coefficients of the feedback function so that the sensor reaches a resonance state, thus reducing the noise interference and improving the sensitivity of the sensor. The simulation results show that optimizing the odd polynomial feedback coefficients with PSO enables the sensor to reach a resonance state, improving sensitivity by at least 23.5%, effectively enhancing sensor performance and laying a foundation for advancements in UAV aeromagnetic measurement technology.

## 1. Introduction

Aeromagnetic gradient exploration enables the rapid and accurate acquisition of the distribution of magnetic mineral resources in the target region. In the terrains of China—characterized by high altitudinal differences, rugged terrain with crisscrossed ravines and dense waterbody distribution—traditional aeromagnetic detection methods fail to obtain geomagnetic distribution data effectively. Due to their characteristics of miniaturization, intelligence, flexibility in transportation and ease of operation, UAVs can maintain low-altitude flights in complex areas, facilitating low-altitude aeromagnetic measurements [1]. In recent years, UAV-based aeromagnetic measurements have emerged as a research focus in geophysical research worldwide [2].

Currently, only superconducting magnetometers and high-sensitivity fluxgate sensors are suitable for geomagnetic vector gradient measurements [3,4]. The fluxgate sensor is characterized by its small size, lightweight design and low manufacturing cost, aligning with the trend of lightweight measurement systems using UAVs as platforms. However, even-harmonic fluxgate sensors face significant interference from odd-harmonic components in their detection principles, and complete symmetry of the dual-axis structure in sensitive units is difficult to achieve, limiting their sensitivity and resolution. The RTD fluxgate sensor utilizes the bidirectional magnetic saturation time difference of the uniaxial magnetic core to measure the magnetic field, which avoids the problems of the even-harmonic type fluxgate from the measurement principle [5]. In order to meet the needs of UAV aeromagnetic gradient detection applications, the sensitivity of RTD fluxgate sensors needs to be further improved.

The magnetic cores of the sensitive units in RTD fluxgate sensors use high-permeability soft magnetic materials, which directly affect the sensor’s sensitivity. However, such high-permeability soft magnetic cores are prone to noise interference. The noise, intrinsic to the magnetic core’s properties, is difficult to eliminate through detection circuits, thereby hindering sensitivity improvements. Thus, it is crucial to investigate the bistable characteristics of magnetic cores and sensor noise to reduce the dependence of sensor sensitivity on the magnetic core’s properties. Given the bistable characteristics of sensitive unit magnetic cores, RTD fluxgate sensors can be approximated as bistable systems. In the study of bistable sensor systems, P. Longhini et al. studied the theory and mathematical model of bistable nonlinear sensors and proposed the fluxgate in the form of coupled arrays [6]; J. L. Aven et al. analyzed the output characteristics of fluxgate constituted by a cascade of three nonlinear sensors in different noise environments, and constructed a model of a multistable system with adjustable parameters using a bistable system [7]. The above studies provide theoretical and practical references for the subsequent studies of bistable sensor systems, but the above methods are limited to the use of cascade to improve the performance of the sensors, and there is a significant lack of research on noise analysis and conversion. Long Changcai et al. proposed to use Gaussian white noise as a driver to establish the relationship between the accumulated time difference of bidirectional saturation of the magnetic core and the measured magnetic field to solve the problem of the measurement accuracy being difficult to improve, but the method used to calculate the time difference is complicated, which introduces a quantization error [8]. Yang Bo et al. proposed to use Gaussian noise and sinusoidal signals to drive RTD fluxgates, which improves the sensitivity, but the actual input/output curves of the sensors are not the same as the theoretical curves [9]. These methods are limited to using noise as a driving mechanism for the sensors, while approaches that enhance sensitivity by adjusting sensor system parameters to modify noise response characteristics have been rarely studied.

Since the RTD fluxgate sensor characteristics are consistent with the basic model of stochastic resonance and satisfy the conditions of the stochastic resonance phenomenon, the sensitivity of the sensor reaches its maximum when a stochastic resonance is generated. Current feedback forms are relatively simple. Linear feedback can alter coercivity, and cubic higher-order feedback can adjust the sensitivity peak. Experimental comparisons in this study reveal that ninth-order feedback yields the best performance. Therefore, this paper proposes an odd-order polynomial feedback form with a maximum ninth-order term, which can simultaneously adjust coercivity and sensitivity peaks.

Due to the reliance on fixed or empirical parameters in existing methods, improving the sensitivity of RTD fluxgate sensors remains challenging. The use of stochastic resonance methods can address this issue by adaptively adjusting feedback parameters, indirectly tuning sensor system parameters. This approach induces stochastic resonance between the excitation signal and input noise, ultimately enhancing the sensor’s sensitivity. The proposed method addresses existing issues such as poor dynamic adaptability, independent design between feedback and stochastic resonance mechanisms, high computational cost and slow convergence of current methods. By dynamically tuning the parameters of the RTD fluxgate to achieve resonance, this method significantly improves its sensitivity.

## 2. Woking Principle of RTD Fluxgate and Stochastic Resonance Theory

### 2.1. Woking Principle of RTD Fluxgate

The RTD fluxgate consists of a sensitive unit and detection circuit; the sensitive unit core uses high-permeability soft magnetic materials because the soft magnetic materials have bistable properties and hysteresis saturation characteristics [10,11] when the periodic alternating excitation magnetic field ${\;H}_{e}(t)$ acting on the magnetic core of the sensitive unit is greater than the coercive force $H_{c}$. The magnetic core reaches the bidirectional saturation state, and the time that the magnetic core stays in the positive and negative saturation state during magnetization will change with the measured magnetic field $H_{x}$ [12,13,14]. The working principle of the RTD fluxgate is shown in Figure 1. When the measured magnetic field $H_{x}$ ≠ 0, the time interval $T^{+}$ between the positive and negative pulses of the output induced voltage of the sensitive unit is different from the time interval $T^{-}$ between the negative and positive pulses. The measured magnetic field $H_{x}$ is measured by detecting the bidirectional magnetic saturation time difference ΔT between the positive and negative pulses of the output sensing voltage.

### 2.2. Stochastic Resonance Theory

Stochastic resonance refers to the cooperative interaction between a nonlinear system, noise signal and input signal [15]. In a nonlinear bistable system, when the input excitation signal is fixed, as the intensity of the noise signal increases, the signal-to-noise ratio (SNR) of the output signal first decreases, then increases until the SNR reaches its maximum value. As the noise intensity continues to increase, the SNR of the system decreases again. This process can be described as stochastic resonance, and the phenomenon where the SNR reaches its maximum value is called the resonance phenomenon [16]. The composition of stochastic resonance is shown in Figure 2.

The potential energy function of the state of the magnetic core in the RTD fluxgate sensor belongs to a bistable function, meaning it has two stable equilibrium points, as shown in Equation (1).(1)$$U\left(x,t\right)=\frac{x^{2}}{2}-Mlncos\left[\frac{x+H_{e}\left(t\right)+\varepsilon }{M}\right]$$

$H_{e}(t)$ represents the excitation magnetic field, *ε* is the measured magnetic field and *M* is a control parameter determined by temperature. According to Equation (1), it is evident that the magnetic core of the RTD fluxgate exhibits bistable characteristics, indicating that the RTD fluxgate possesses the features of the bistable system. Furthermore, the input to the RTD fluxgate which consists of a fixed-frequency excitation signal and the presence of input noise enables the system to meet the conditions for stochastic resonance and has the condition to generate stochastic resonance.

## 3. Research on Adaptive Feedback Control of RTD Fluxgate Based on Stochastic Resonance

### 3.1. Study on Feedback Structure of RTD Fluxgate Based on Stochastic Resonance

The sensitivity of the RTD fluxgate is closely related to the time difference ΔT and the measured magnetic field *H*_x_. The sensitivity S of the RTD fluxgate is defined as the ratio of the time difference to the measured magnetic field $H_{x}$:(2)$$S=\frac{\Delta T}{H_{x}}$$

From Equation (2), it can be seen that when the measured magnetic field $H_{x}$ is constant, the sensitivity S is directly proportional to the time difference ΔT. Therefore, to enhance the sensitivity while keeping the measured magnetic field unchanged, ΔT must be increased. However, it is challenging for the RTD fluxgate to achieve a resonant state by directly altering system parameters. Appropriate feedback functions can be designed to indirectly adjust the system parameters, thereby enabling the system to reach resonance and maximize the time difference, thus improving sensitivity. The bistable characteristics of the RTD fluxgate can be described by the Langevin equation, as shown in Equation (3).(3)$$\frac{dx}{dt}=-\frac{dV}{dx}+u\left(t\right)$$

$u\left(t\right)=s\left(t\right)+\xi \left(t\right)$, $s\left(t\right)$ is the input signal and $\xi \left(t\right)$ is the Gaussian white noise signal. $V\left(x\right)=-\frac{1}{2}ax^{2}+\frac{1}{4}bx^{4}$ and *a* and *b* are positive constants. After derivation of the equation, it exhibits distinct odd-order characteristics, leading to the conclusion that the bistable system possesses features of odd-order functions. Because the selection of the feedback form can directly affect the sensitivity of the RTD fluxgate, this study selects an odd-order feedback structure based on the bistable characteristics, specifically exploring linear, cubic and higher-order odd feedback forms to discuss the relationship between the time difference and noise intensity.

(1)When the linear feedback is introduced, with the feedback term represented as *k*_1_y(*t*), the corresponding equation for the simulation model is



(4)
$$y\left(t+\Delta t\right)=sgn\left[\left(K+k_{1}\right)y\left(t\right)+s\left(t\right)+\xi \left(t\right)\right]$$



Let *K*’ = *K* + *k*_1_, then the equation simplifies to(5)$$y\left(t+\Delta t\right)=sgn\left[K^{'}y\left(t\right)+s\left(t\right)+\xi \left(t\right)\right]$$

The linear feedback coefficient *k*_1_ can be simplified and integrated into the overall feedback coefficient *K*. The primary role of linear feedback is to adjust the coercive force of the magnetic core. When the excitation amplitude and the measured magnetic field remain constant, the RTD fluxgate, utilizing a linear feedback structure, can indirectly regulate the coercive force of the magnetic core, although its effect on noise response characteristics is minimal. Research indicates that reducing the coercive force can enhance the sensitivity of the RTD fluxgate by decreasing the excitation amplitude [17]. Therefore, the feedback coefficient is taken as the variable of the RTD fluxgate system, and the sensitivity is optimized by comparing the time difference under different feedback coefficients, adjusting the feedback coefficient *k*_1_ and changing the excitation amplitude A. The relationship between the time difference ΔT and noise intensity is illustrated in Figure 3.

From Figure 3, it can be observed that as the noise intensity increases, the time difference ΔT initially decreases, then rises, reaching a peak before continuously declining. With the increase in the feedback coefficient and excitation amplitude, the peak value of time difference gradually decreases, while the corresponding noise intensity at the peak value gradually increases. Linear feedback can adjust the coercive force of the magnetic core. Thus, the sensitivity can be improved by reducing the excitation amplitude.

(2)When odd higher-order feedback is introduced, with the feedback term represented as *k*_2n+1_y^2n+1^, the corresponding equation for the simulation model is



(6)
$$y\left(t+\Delta t\right)=sgn\left[Ky\left(t\right)+k_{3}y^{3}\left(t\right)+\cdots +k_{2n+1}y^{2n+1}\left(t\right)+s\left(t\right)+\xi \left(t\right)\right]\left(n>0\right)$$



As shown in Figure 4, the sensitivity is controlled by adjusting the feedback function coefficient to modify the optimal noise for the RTD fluxgate. Under different noise intensities, the shape of the noise response characteristic curve remains largely consistent. When using the higher-order odd feedback structures, the RTD fluxgate can achieve maximum sensitivity within a certain range of noise; the cubic feedback can adjust the peak position of sensitivity, allowing it to actively adapt to the external noise.

From Figure 4, it can be observed that with the increase in noise intensity, each graph exhibits a peak value. As the feedback coefficient increases, the numerical value of the time difference ΔT peak changes minimally. However, the corresponding noise intensity at the time difference ΔT peak continuously increases. This indicates that the odd-order feedback can effectively adjust the maximum sensitivity.

In summary, this paper proposes a structure combining linear and high-order feedback, which can effectively adjust the coercive force and sensitivity peak position, enhance the sensitivity of the RTD fluxgate and reduce the impact of noise under uncertain noise characteristics. However, it is practically challenging to manually adjust the feedback parameters one by one to achieve stochastic resonance. However, current optimization methods are largely based on static experimental scenarios, lacking dynamic adaptability. Moreover, the feedback design and stochastic resonance control are independent of each other, making joint optimization difficult to achieve. Thus, this paper introduces the PSO algorithm to dynamically calculate the odd-order polynomial coefficients, achieving a resonance state through an adaptive method, indirectly improving the sensor sensitivity.

### 3.2. Adaptive Algorithm of RTD Fluxgate with the Feedback Coefficient of Stochastic Resonance

The PSO algorithm simulates the behavior of particles within a group to find the optimal solution to the objective function, and its objective function serves as the criterion for evaluating the quality of each particles’ position during the optimization process [18,19]. The PSO algorithm possesses the ability of global optimization; this paper studies the odd feedback form subject to multiple feedback coefficients *k*_1_–*k*_9_ which can only use the PSO algorithm for a global search to lock the optimal solution. The optimized PSO algorithm objective function is in the form of y = *k*_9_x^9^ + *k*_7_x^7^ + *k*_5_x^5^ + *k*_3_x^3^ + *k*_1_x. The optimal solution of the output refers to the feedback coefficients *k*_1_–*k*_9_ of the dynamic output, so that output correlation of the RTD fluxgate is at its maximum to achieve a resonance state and improve the sensitivity of the RTD fluxgate. The PSO algorithm uses iterative optimization to gradually optimize the feedback coefficients of the RTD fluxgate to improve its sensitivity. The performance of the PSO algorithm can be optimized by adjusting the objective function, and the required feedback function is in the form of an odd polynomial. Therefore, the objective function is set in the form of an odd-degree polynomial to better accommodate the solution of the feedback function coefficient discussed in this paper. The PSO algorithm is applied to the solution of the feedback function coefficients to optimize multiple feedback function coefficients *k*_1_ to *k*_2n+1_; at this time, the particles represent the values of the feedback coefficients which require solving, and the problem at hand involves dynamically adjusting the combination of the feedback function coefficients using the PSO algorithm [20,21]. The algorithm consists of a population of *N* feedback coefficient particles flying at a certain speed in a *D*-dimensional search space, updating the individual extreme value and global extreme value of each feedback coefficient according to the fitness value of each feedback coefficient *k*_1_ to *k*_2n+1_. During the iteration process, the position of the i-th feedback coefficient particle is *X_i_* = (*x_i_*_1_, *x_i_*_2_,..., *x_iD_*)^T^, representing a point in the algorithm’s search space, and the velocity is *V_i_* = (*v_i_*_1_, *v_i_*_2_,..., *v_iD_*)^T^. The individual optimal position of a particle during the entire process is *Pbest_i_* = (*pbest_i_*_1_, *pbest_i_*_2_,..., *pbest_iD_*)^T^, and the global optimal position of all particles found during the search is *Gbest* = (*gbest_i_*_1_, *gbest_i_*_2_,..., *gbest_iD_*)^T^ [22,23,24,25], corresponding to the optimal point for the feedback function coefficient values in the search space. The velocity and position of the feedback coefficient particles are updated according to the following equations [26]:(7)$$v_{id}^{k+1}=\omega v_{id}^{k}+c_{1}r_{1}\left[{pbest}_{id}^{k}-x_{id}^{k}+c_{2}r_{2}\left({gbest}_{id}^{k}-x_{id}^{k}\right)\right]$$(8)$$x_{id}^{k+1}=x_{id}^{k}+v_{id}^{k+1}$$
where *i*∈{1,...,*N*}, *d*∈{1,...,*D*}; *k* is the current iteration number; *c*_1_ is the individual learning factor; and *c*_2_ is the social learning factor, controlling both the particles’ ability to learn from the individual best position (*Pbest*) and the global best position (*Gbest*). Their values should not be too large, as this may cause the particles to converge too early, weakening their ability to escape from the global optimum, and typically, these values are set between 1.5 and 2.5, so in this paper, we set *c*_1_ = *c*_2_ = 2. *r*_1_ and *r*_2_ are random numbers uniformly distributed between [0, 1], and *D* is the dimensionality of the search space [27,28,29,30,31]. *ω* is the inertia weight, adaptively adjusted according to the population’s search state, as shown in Equations (9)–(11).
(9)$$\omega =\omega_{max}\frac{\Delta f}{f_{avg}}$$(10)$$f_{avg}=\frac{1}{N}\sum_{i=1}^{N}f\left(X_{i}\right)$$(11)$$\Delta f=\sqrt{\frac{1}{N}\sum_{i=1}^{N}{\left(f\left(X_{i}\right)-f_{avg}\right)}^{2}}$$

The standard deviation of the current population’s fitness values  Δf is used to measure the population’s diversity. *f_avg_* is the average fitness value of the population; *f*(*X_i_*) is the fitness value of particle *i*; and *N* is the number of particles. The process for solving the feedback coefficients using the PSO algorithm is illustrated in Figure 5.

By utilizing the PSO algorithm to determine the optimal coefficients of the feedback control in the RTD fluxgate, the impact of using optimal feedback coefficients of feedback control on improving RTD fluxgate sensitivity can be assessed based on the fitness function values. The fitness function is designed to evaluate a potential feasible solution, namely the optimal value of the feedback coefficients [32]. In this study, the mean square error is adopted as the fitness function to quantify the discrepancy between the model’s predicted values and actual values. When the fitness value reaches a certain threshold, the time difference is maximized and the RTD fluxgate attains its optimal resonance state. The output value from the PSO algorithm corresponds to the best combination of feedback coefficients, as described by Equation (12).(12)$$fitness=\sum {\left(function(x,coeffs)-function(x,target)\right)}^{2}/n$$
where ‘fitness’ represents the fitness value; ‘function’ denotes the feedback function given by y = *k*_9_x^9^ + *k*_7_x^7^ + *k*_5_x^5^ + *k*_3_x^3^ + *k*_1_x; ‘coeffs’ indicates the model’s predicted values which are the coefficients of the actual output of the feedback function; ‘target’ refers to the target function value, representing the desired combination of function coefficients; and ‘n’ denotes the number of feedback coefficient particles.

## 4. Simulation Data Analysis

To validate the effectiveness of the proposed method, this study utilizes MATLAB (R2022a) Simulink to construct the RTD fluxgate model, as illustrated in Figure 6.

### 4.1. Construction and Output of the Simulation Model

The excitation magnetic field $H_{e}(t)$ is generated using a signal generator, using a sinusoidal waveform as the input excitation signal, and the excitation signal amplitude is set to 1 A/m, the frequency is 1 Hz and the sampling time is 0.01. In order to enable the RTD fluxgate to achieve the simulation of the impact of the noise in the real environment, the noise signal is generated using a white noise module, and the parameters such as the noise frequency and sampling time are set. The measured magnetic field $\;H_{x}$ is generated using a constant generator, and changing the value of this constant can directly adjust the size and direction of the measured magnetic field. The adder adds the excitation magnetic field $H_{e}(t)$, the measured magnetic field ${\;H}_{x}$ and the white noise to obtain the external magnetic field $h\left(t\right)$, which will be passed as an input to the subsequent delay module. The delay module converts the input sinusoidal waveform into a rectangular waveform with an easily observable time difference by setting the values of the turn-on point, turn-off point, output at turn-on and output at turn-off parameters. The feedback control section consisting of the MATLAB function module takes the form of y = *k*_9_x^9^ + *k*_7_x^7^ + *k*_5_x^5^ + *k*_3_x^3^ + *k*_1_x as the feedback function. A multiplication module and a differentiation module work together to generate the induced voltage $e\left(t\right)$, while a counter module counts the induced voltage $e\left(t\right)$, leading to a pulse graph and calculating the time difference ΔT between pulses. The simulation model outputs are illustrated in Figure 7 and Figure 8.

### 4.2. Comparison of Algorithms Performance for Different Form of Feedback

#### 4.2.1. Linear Feedback

In order to verify the effectiveness of the PSO algorithm in terms of the linear feedback function, it is compared with the fixed-step single-objective algorithm. Since the fixed-step single-objective algorithm only optimizes a single parameter, it cannot dynamically solve for the best combination of multiple feedback coefficients, and its algorithmic step size is pre-set and remains unchanged during execution. The PSO belongs to the multi-objective optimization algorithm, which not only can simultaneously find out the most suitable feedback parameters in the global scope, but can also ensure the correlation between parameters, improve the accuracy of optimized parameters and shorten the search time, and its relation to the fixed-step single-objective algorithm is more suitable for solving the linear feedback coefficients of the RTD fluxgate.

#### 4.2.2. Odd-Order Feedback

In order to verify the effectiveness of the PSO algorithm in solving the optimal odd-order coefficients of the odd-order feedback function, this paper compares the PSO algorithm with the Gray Wolf Optimizer (GWO). The GWO is an intelligent optimization algorithm inspired by the leadership strategy and hunting behavior of the gray wolf which can avoid falling into local optimal solutions [33,34,35]. Based on the parameters of Figure 6, the optimal coefficient combination of the feedback function is selected with an adaptation degree between 0.12 and 0.93 to compare the time difference between the two algorithms, as shown in Table 1 and Figure 9.

The comparison of the above table shows that the feedback parameters solved by the PSO algorithm can be applied to the RTD fluxgate to obtain a larger time difference and a relatively higher sensitivity in the case of the two algorithms with the same range of fitness. With the GWO, the optimal time shown in Equation (9) is 0.78 when the fitness is 0.19, while the optimal time difference is 1.11 when the fitness is 0.15 with the PSO algorithm. The PSO algorithm improves the optimal time difference by 42.31% compared with the GWO. From the above figure, it can be seen that the PSO algorithm is more suitable for solving the odd-order feedback coefficients of the RTD fluxgate compared to the GWO.

#### 4.2.3. Odd-Order Polynomial Feedback

In order to verify the effectiveness of the proposed method in this paper in terms of the odd-order polynomial feedback functions, the PSO algorithm was compared with the Ant Colony Optimization (ACO) algorithm, which can also solve multiple parameters. The ACO algorithm is an optimization algorithm based on the abstract modeling of ants’ food-seeking behavior. This algorithm is capable of finding optimal solutions for multiple parameters and evaluates the quality of output solutions based on fitness. Based on the parameters in Figure 6, optimal feedback function coefficient combinations were selected with fitness values ranging from 0.04 to 1.60 to compare the time differences between the two algorithms, as shown in Table 2 and Figure 10.

The comparison indicates that both ACO and PSO are suitable for solving odd-order polynomial feedback functions in the RTD fluxgate, allowing for the determination of optimal feedback coefficient combinations. But when the ACO algorithm achieves a fitness value of 0.5, the maximum time difference which reaches 1.11 is lower than the time difference of the PSO algorithm. In contrast, the maximum time difference obtained using the PSO algorithm is 1.5, representing a 32.4% improvement over the time difference achieved with the ACO algorithm.

### 4.3. Comparison of Outputs Using PSO for Linear and Odd-Order High-Order Feedback Forms

To investigate the correlation between the time difference Δ*T* when different feedback forms are used, this paper carries out several sets of experiments by changing the coefficient *k* of the feedback function. When the amplitude and frequency of the excitation signal and the measured magnetic field are fixed, the paper compares the output time difference of the RTD fluxgate under different feedback functions to study the optimal feedback form for each performance index, and the results of the experiments are shown in Figure 11.

Figure 11 shows that when the feedback function is set to y = *k*_9_x^9^, it exhibits a larger time difference compared to other lower-order odd functions. When the higher-order odd feedback form is adjusted to y = *k*_11_x^11^, the increment in the time difference is relatively small. With the feedback function set to y = *k*_13_x^13^, there is almost no increase in the time difference. Therefore, increasing the order does not significantly change the increment of time difference; thus, the feedback function in the ninth power form yields the maximum time difference. However, using y = *k*_9_x^9^ as the feedback function does not result in a significant change in the output time difference of the RTD fluxgate, and the trend remains stable. Consequently, a comparison of the output time differences was made between the odd-order polynomial feedback function and y = *k*_9_x^9^, as shown in Figure 12.

Figure 12 shows that when the feedback coefficients are all set to 0.2, the time difference obtained using the feedback function y = *k*_9_x^9^ is 0.55, while the time difference for the feedback function y = *k*_9_x^9^ + *k*_7_x^7^ + *k*_5_x^5^ + *k*_3_x^3^ + *k*_1_x is 0.66, indicating a relative sensitivity improvement of 20%. Consequently, the time difference values of odd-order polynomial feedback functions are consistently greater than those of y = *k*_9_x^9^. Therefore, selecting an odd-order polynomial form as the feedback function can effectively enhance sensitivity. Subsequently, the feedback coefficients *k*_1_ to *k*_9_ derived from the PSO algorithm were incorporated into the function module of the simulation model. The performance of the obtained feedback parameters was assessed based on the fitness function values, and the relationship between the time difference calculated from the pulse interval differences and the fitness values is presented in Table 3 and Figure 13.

The results indicate that applying random resonance theory and feedback control to RTD fluxgates can enhance their sensitivity. By using the PSO algorithm to simulate the search for optimal particles’ positions and movement directions, the optimal feedback control coefficients can be determined. As the feedback function coefficients are adjusted, the optimal feedback coefficients can be obtained, thereby maximizing time difference, which leads the RTD fluxgate to reach a state of random resonance. The computational cost of the GWO, ACO and PSO algorithms per iteration is *O* (*P* × *N*), where *P* is the population size and *N* is the problem dimension. The total computational cost is *O* (*T*_max_ × *P* × *N*), which represents the maximum number of iterations and which affects the overall computation. If the change in the optimal solution during *N* iterations is less than a threshold *ε_best*, the algorithm is considered to have converged. As shown in Figure 13, the fitness starts from zero and gradually increases, while the time difference decreases. When the fitness reaches a certain value, the time difference output of the RTD fluxgate attains its maximum. If the fitness continues to increase, the time difference further decreases. Therefore, when the fitness is 0.35, the RTD fluxgate reaches resonance, yielding a time difference of 1.4, which is a 23.5% improvement compared to the time difference of the non-feedback model with sensitivity achieving its maximum value. The results demonstrate that the integration of random resonance theory and feedback control in the RTD fluxgate effectively improves sensitivity. By optimizing the feedback control coefficients through the PSO algorithm, the coefficients that maximize the time difference are obtained, allowing the system to reach the state of random resonance. As shown in Figure 13, when the fitness reaches 0.35, the time difference of the RTD fluxgate is 1.4, reflecting a 23.5% enhancement over the original model, with sensitivity at its peak.

## 5. Conclusions

In this paper, the idea that the core of the RTD fluxgate unit has bistable characteristics and hysteresis saturation characteristics is elaborated upon. Based on the study of the excitation signal and input noise characteristics, it is proposed that we realize the stochastic resonance of the sensor system by adding feedback. The system parameters of the RTD fluxgate are adjusted indirectly by introducing odd polynomial feedback, and the optimal coefficients of the odd polynomials are obtained by combining the PSO algorithm with an iterative optimization search, so that the RTD fluxgate can reach a resonance state. The simulation result shows that when the sensor resonates with the feedback coefficients obtained by the optimized PSO algorithm, the sensitivity is improved by 23.5% relative to the case of no feedback function, by 32.43% compared to the case of optimizing the feedback coefficients using the GWO and by 42.31% compared to the case of optimizing the feedback coefficients using ACO. The comparison shows that the method proposed in this paper significantly improves the sensitivity of the RTD fluxgate and reduces the effect of random noise on the sensitivity. The success of this study expands the new fluxgate detection principle and research direction and positively promotes fluxgate measurement accuracy and stability.

## Figures and Tables

**Figure 1 sensors-25-00520-f001:**
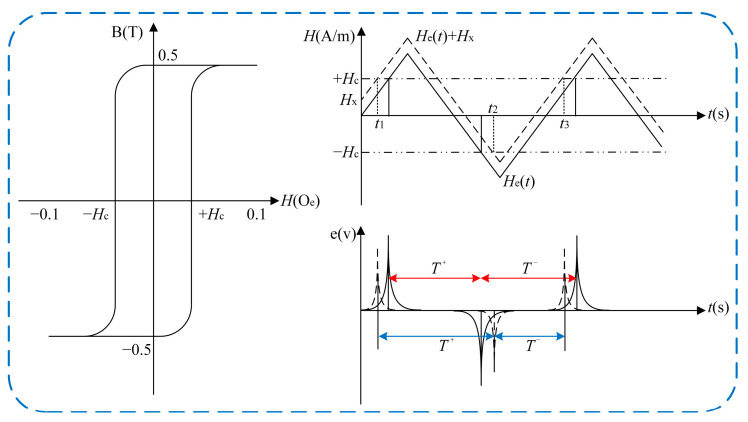
Working principle of RTD fluxgate based on bistable characteristics of magnetic core hysteresis saturation.

**Figure 2 sensors-25-00520-f002:**
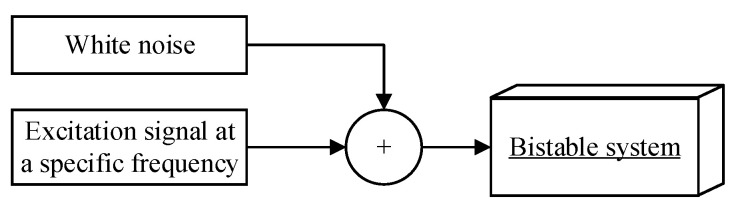
A diagram of stochastic resonance components.

**Figure 3 sensors-25-00520-f003:**
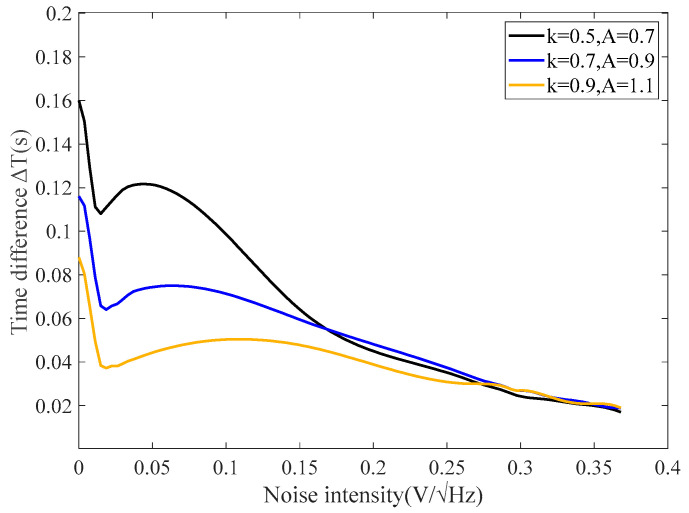
Relationship between time delay and noise intensity with varying coercive force.

**Figure 4 sensors-25-00520-f004:**
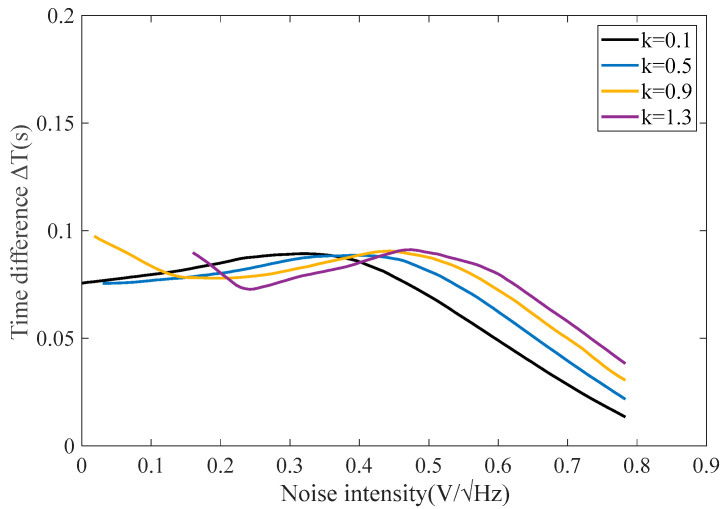
Stochastic resonance curves of RTD fluxgate with different cubic feedback coefficients.

**Figure 5 sensors-25-00520-f005:**
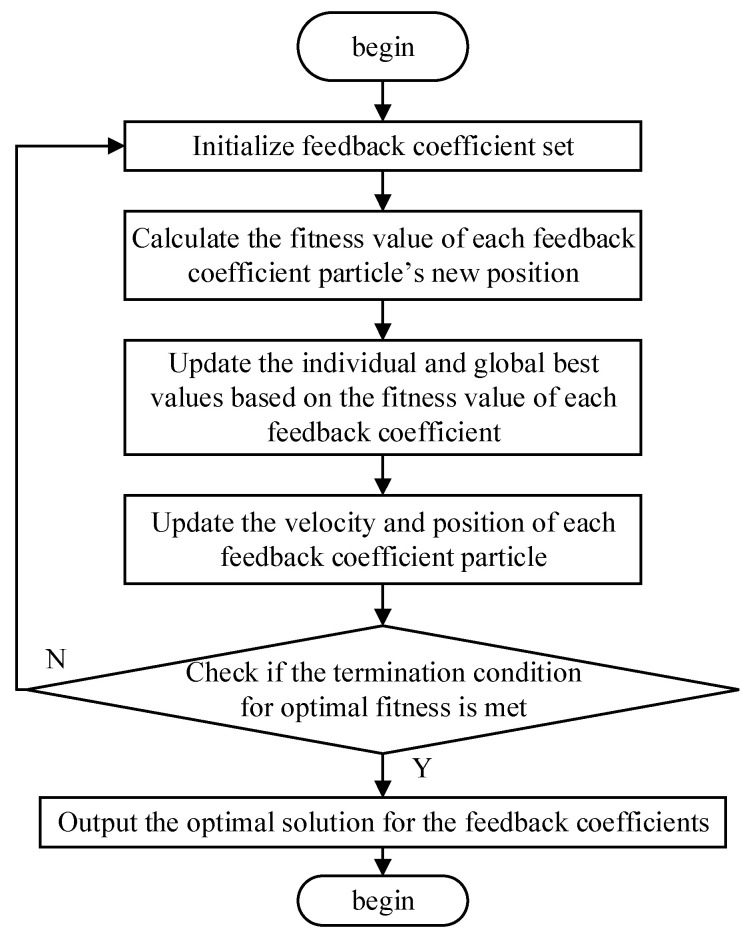
The flowchart of the PSO algorithm.

**Figure 6 sensors-25-00520-f006:**
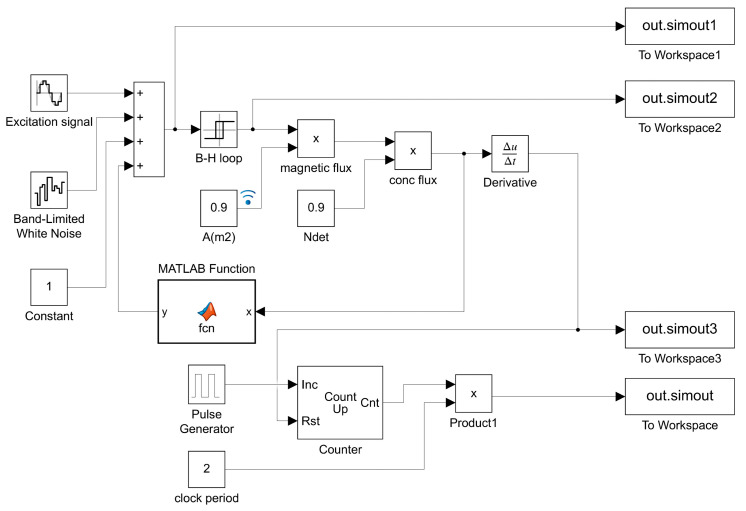
Simulation model of RTD fluxgate with feedback.

**Figure 7 sensors-25-00520-f007:**
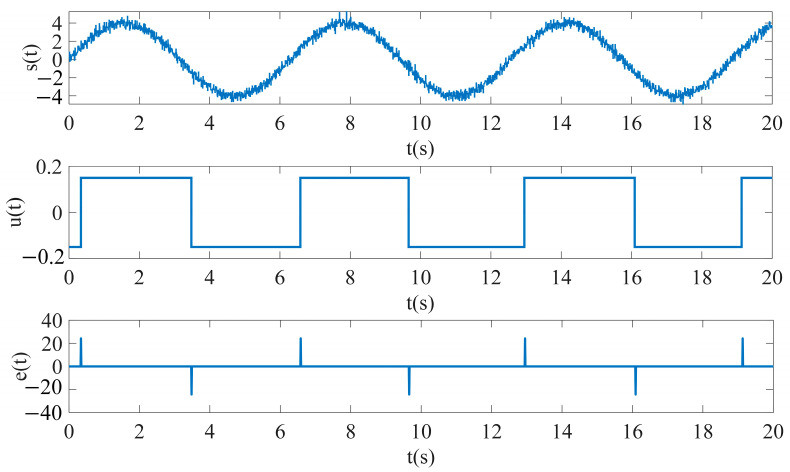
Simulation results of RTD fluxgate (*H*_x_ = 0).

**Figure 8 sensors-25-00520-f008:**
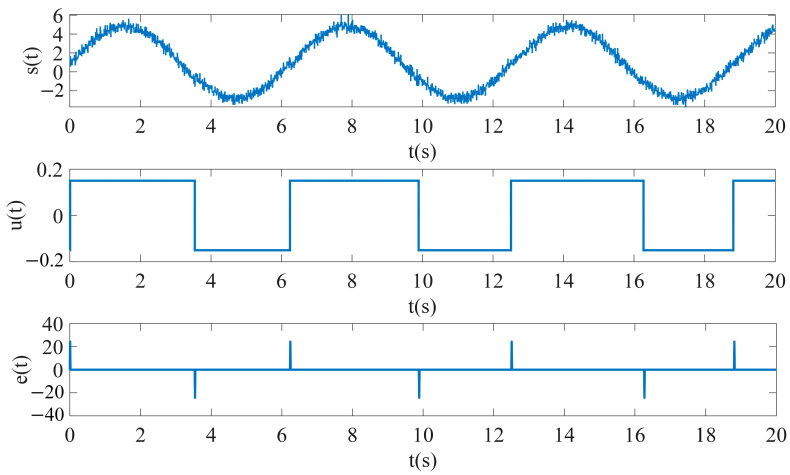
Simulation results of RTD fluxgate (*H*_x_ ≠ 0).

**Figure 9 sensors-25-00520-f009:**
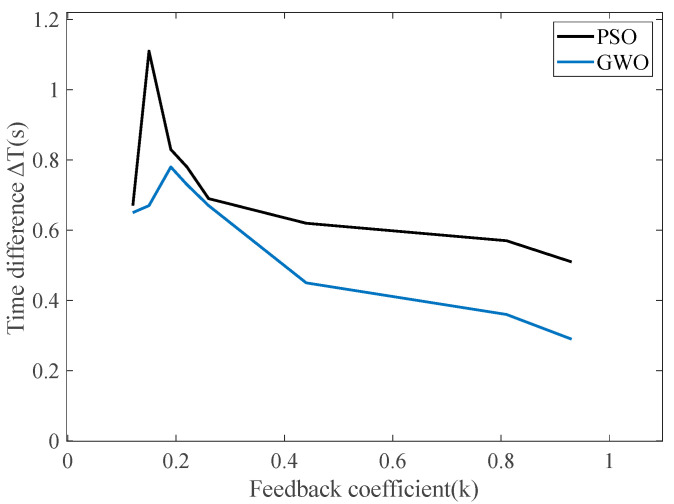
Comparison of time differences between GWO and PSO.

**Figure 10 sensors-25-00520-f010:**
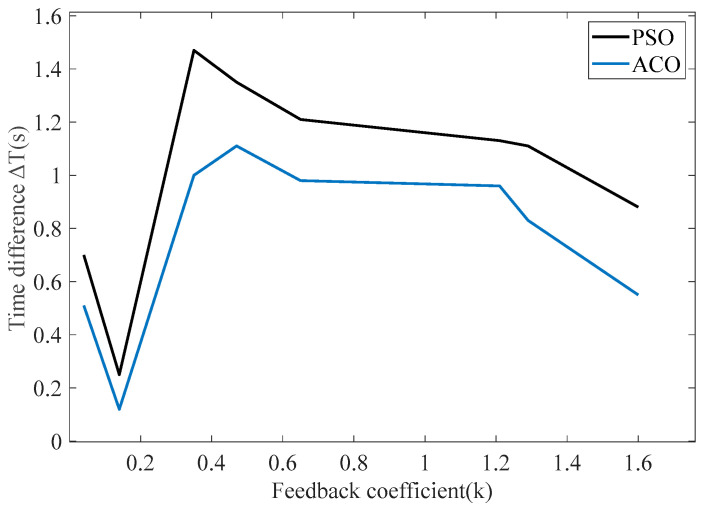
Relationship between fitness and time difference obtained by ACO.

**Figure 11 sensors-25-00520-f011:**
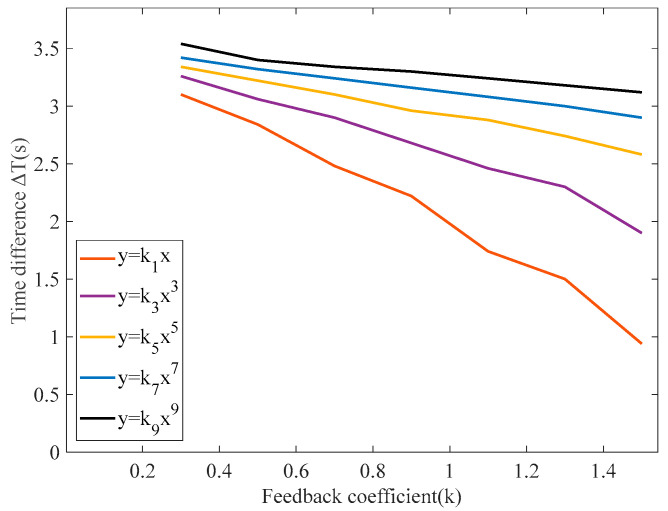
Comparison of output time delay under different feedback functions.

**Figure 12 sensors-25-00520-f012:**
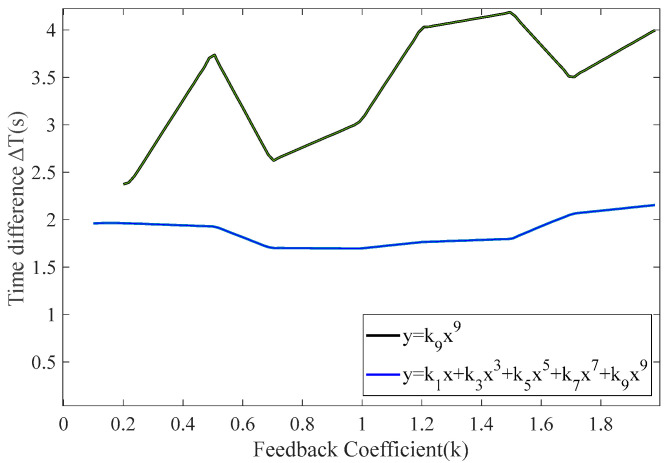
Comparison between y = *k*_9_x^9^ and y = *k*_9_x^9^ + *k*_7_x^7^ + *k*_5_x^5^ + *k*_3_x^3^ + *k*_1_x.

**Figure 13 sensors-25-00520-f013:**
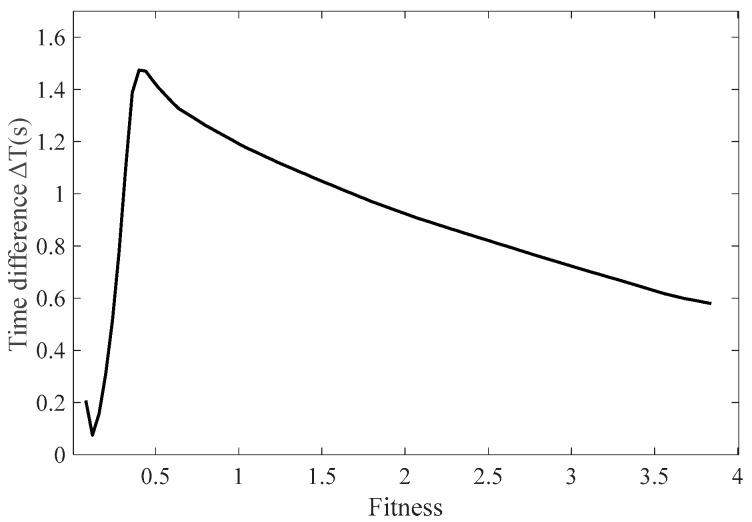
Relationship between fitness and time difference for different feedback functions.

**Table 1 sensors-25-00520-t001:** Relationship between fitness and time difference obtained by GWO.

Fitness	Time Difference in PSO	Time Difference in GWO
0.12	0.67	0.65
0.15	1.11	0.67
0.19	0.83	0.78
0.22	0.78	0.73
0.26	0.69	0.67
0.44	0.62	0.45
0.81	0.57	0.36
0.93	0.51	0.29

**Table 2 sensors-25-00520-t002:** Relationship between fitness and time difference obtained by ACO.

Fitness	Time Difference in PSO	Time Difference in ACO
0.04	0.7	0.51
0.14	0.25	0.12
0.35	1.47	1.00
0.47	1.35	1.11
0.65	1.21	0.98
1.21	1.13	0.96
1.29	1.11	0.83
1.60	0.88	0.55

**Table 3 sensors-25-00520-t003:** Relationship between fitness and time difference for different feedback functions.

Fitness	Time Difference
0.04	0.7
0.14	0.25
0.25	0.42
0.32	0.45
0.35	1.47
0.65	1.21
1.21	1.13
1.29	1.11
1.60	0.88
2.71	0.61
3.74	0.49

## Data Availability

Data are contained within the article.

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
