# Peer review of "Research on Sensitivity Improvement Methods for RTD Fluxgates Based on Feedback-Driven Stochastic Resonance with PSO"

_sensors, 2025, doi:10.3390/s25020520_

Round 1
Reviewer 1 Report
Comments and Suggestions for Authors
1. The introduction outlines the contributions of other research institutions to improve the sensitivity of RTD fluxgate sensors, but does not point out the problems with the current research status. The introduction needs to be significantly revised to highlight the importance and value of this research.
2. The PSO algorithm is prone to falling into local optimal solutions. Does the manuscript need to consider the occurrence of this situation when using this algorithm?
3. The introduction of the adaptive algorithm of RTD-fluxgate with the feedback coefficient stochastic resonance is not clear enough. It only briefly explains the principle of the PSO algorithm, and does not analyze the specific problems raised in the manuscript. In what dimension is the problem? What does the position ‘Xi’ refer to in this problem? What exactly is the optimal output? That is, what does ‘Gbest’ refer to in this problem? The objective function is not explained clearly enough.
4. Is it possible to add relevant experiments to verify the simulation results?
5. There seems to be something wrong with the format of Table 3.
6. The quality of the images in the manuscript needs to be improved, especially Figures 1, 9, and 10. Although other images are not directly pointed out, the images are not clear enough.
Comments on the Quality of English LanguageYour English prose mostly is fine, and certainly communicates what is critical, but there are a number of cases of misuse of articles or strange phrases. It would be good if you have a native English-speaking colleague to read over the manuscript and suggest minor changes that make the article more readable.
Reviewer 2 Report
Comments and Suggestions for Authors
“Review on Sensitivity Improvement Methods for RTD-Fluxgates Based on Stochastic Resonance.” It is an important topic in enhancing RTD-fluxgate sensor sensitivity for UAV aeromagnetic surveys, and the work is clearly structured. However, before it can be considered for publication, there are several key areas that need attention.
Here are my main concerns and suggestions:
*title suggests that the paper is a "Review," but it appears to introduce a new method for improving sensitivity using feedback-driven stochastic resonance with PSO. If it's primarily a research paper presenting novel findings, consider revising the title to better reflect that.
*While the manuscript introduction provides a good background on UAV aeromagnetic surveys and RTD-fluxgate sensors, it doesn't clearly state the gap study is addressing. It would be good to elaborate on how the feedback and optimization strategy differs from existing methods. Highlighting the unique aspects of approach, clarify its novelty and significance.
*The manuscript lacks sufficient details about the PSO and other algorithms (GWO, ACO) used for comparison. It would be better to include specifics on PSO parameters like inertia weight, learning factors, and how determined these settings. Additionally, discussing the computational cost or convergence criteria for each algorithm.
*results are obtained through simulation. Autohers considered experimental validation, or planning to test these feedback strategies on real hardware in the future? Mentioning potential future steps such as experimental trials or exploring more complex feedback functions could provide a stronger conclusion.
*figures are nice, but some captions could be more descriptive, so without having to refer back to the text extensively.
Comments on the Quality of English Language
* language edits for clarity such as: 2. The Woking Principle of RTD-fluxgate and Stochastic Resonance Theory 2.1 The Woking Principle of RTD-fluxgate. Please check spelling throughout the paper.
Round 2
Reviewer 1 Report
Comments and Suggestions for Authors
Thanks to the Authors for addressing my comments, I hope those were useful. From my point of view I see many improvements, but there are still some aspects that could be better addressed as reported in the following.
1. If additional verification experiments can be performed, the effectiveness of the method will be more convincing.
2. Actually figures quality could be still enhanced by using vectorial formats like .eps
3. It is recommended to polish the English expression so that readers can better understand it.
Reviewer 2 Report
Comments and Suggestions for Authors
After the Authors' response to my previous reviews, I recommend the revised manuscript for publication. The conclusions are original and offer a new perspective.
Author Response
Thank you very much for your patient guidance and support for my article. Your suggestions have significantly improved my manuscript. Once again, I sincerely appreciate your recognition of my article. Wishing you success in your work and happiness in life!